# The Story So Far on Narrative Planning

**Primary Keywords:** *(1) Applications;*

## Abstract

Narrative planning is the use of automated planning to construct, communicate, and understand stories, a form of information to which human cognition and enaction is predisposed. We review the narrative planning problem in a manner suitable as an introduction to the area, survey different plan-based methodologies and affordances for reasoning about narrative, and discuss open challenges relevant to the broader AI community.

## 1 The Role of Narrative

The field of artificial intelligence has grappled with modeling story reasoning since its beginning (McCarthy 1990), due in part because the ability to understand and tell stories is thought to underlie or inevitably result from human cognition (Winston 2011).

Research on *narrative intelligence* progresses along three efforts that encompass a multitude of intertwined natural language, commonsense, and multi-agent reasoning tasks: narrative construction, communication, and understanding (Mueller 2013). Common across these efforts is the use of *automated planning* as a formal, rigorous, and common vocabulary for framing advances in the field. This is because AI planning naturally reasons over concepts (*e.g.* agents, objects, states, events) central to plot structure and its communication (Young 1999).

While not all narrative intelligence research *focuses* on planning, it is predominantly plan-based or plan-like. Applications include efforts to model human cognition (Cardona-Rivera et al. 2016), achieve human-level performance on language processing tasks (Martin et al. 2018), demonstrate independent creativity (Summerville et al. 2017), structure human-computer interaction (Porteous, Cavazza, and Charles 2010), and explain AI rationales (Riedl 2016). Strikingly, AI research that does *not* focus on *narrative planning* is re-discovering the utility of data structures and algorithms that form its basis. For example, neuro-symbolic systems that use narrative representations outperform non-trivial baselines in commonsense reasoning (Bosselut et al. 2019; Cohen 2020).

We thus feel the time is ripe to take stock of the state-of-the-art in narrative planning. This research community has converged upon representational and reasoning commitments necessary to account for storytelling and story understanding in people (Cardona-Rivera and Young 2019; Hayton et al. 2020). These commitments are important to survey for advancing AI that can perform as robustly and flexibly as humans do. At the same time, this area is relevant where we already see the use of narrative, when having greater predictive control over narrative effects would benefit society; e.g., in narratives for personalized learning (Wang et al. 2017), rehabilitation therapy and healthcare communication (Ozer et al. 2020), and intelligence analysis (Lukin and Eum 2023). And while there is no established standard for specifying narrative planning problems (Hayton et al. 2020; Shirvani and Ware 2020) there is a vibrant community of practice, whose advances and unsolved problems may bear relevance to the broader AI community.

One work in recent history has surveyed narrative planning (Young et al. 2013), but its contours are imprecise in technical detail. In contrast, we survey the field in formal depth that is sufficient to precisely describe common themes. We structure our survey as a tour guided by the motivating question:

*How might a computer system tell a story?*

The answer is deceptively trite: *it depends*. In unpacking this answer, we illustrate why narrative planning is so vast. Along the way, we use an extended case example to identify where our community has converged and diverged. As we discuss later, the points of convergence are centered on elements deemed necessary for a computer-generated story to be perceived *as a story* by an audience. In turn, the points of divergence concern what *layers of narrative detail* are needed, what *narrative effects* are desirable, and what *planning approaches* are used for modeling.

As a consequence of our exploration, we will cut across narrative planning systems that address the tasks of narrative understanding, construction, and communication. Finally, we briefly sketch promising open problems that might stimulate the broader community's pursuit of new directions.

## 2 Pre-rigorous Notions in an Example

Before we introduce this area in technical depth, we first present more-intuitive descriptions of key narratological concepts. For this, we introduce a running example that makes our motivating question more specific: *how might a computer system tell a story like the one in Figure 1*?

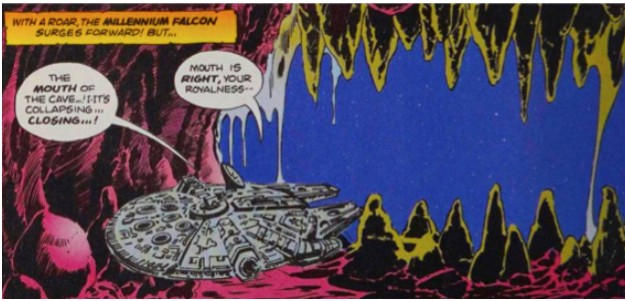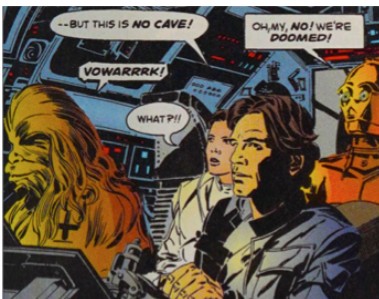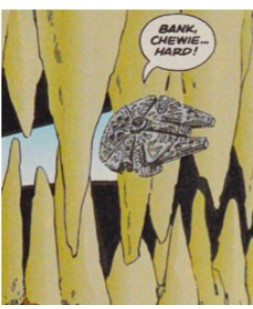

Figure 1: Panels from Marvel's *Star Wars* #39–#44, which depict *Princess Leia*, *Han Solo*, *Chewbacca*, and *C-3P0* realizing they are in the belly of a giant space worm. The crew originally piloted the *Millennium Falcon* into the beast, thinking it was an asteroid cave where they could hide from the relentlessly-pursuing *Empire*. In this paper, we explore how an automated planning system would be able to tell such a story.

The depicted excerpt is from *Star Wars #39–#44* (Goodwin, Williamson, and Garzon 1977), a Marvel Comics adaptation of the film *Star Wars Episode V: The Empire Strikes Back*. Despite its brevity, it is rich enough to discuss *narrative* in two senses that narrative planning models must contend with: as a *communicative act* and as a *designed artifact*.

## 2.1 Narrative as a Communicative Act

Figure 1 reflects an implicit communicative choice: the choice of *narration*—the narrative's surface form realization in a medium (Hühn and Sommer 2013). While we use a comic, we could have used a film, book, or video game.

Audiences familiar with the comic's source film may note differences in content beyond narration; namely, in the *plot*—the narrative's virtual world (Rimmon-Kenan 2002). The comic differs from the film in its *events* (causally and purposively related changes in states-of-affairs). To wit, there are differences in the dialogue of *characters* (anthropomorphized intention-driven agents), and event *locations* (spatial context).

Plot differences reflect why the choice of narration is more than a matter of aesthetic taste: the medium constrains what plot information *can* be narrated (Elliott 2004). Above, the constraints are from the *comic form*; i.e., syntax and semantics of visual language (Martens, Cardona-Rivera, and Cohn 2020). This explains why the last panel contains dialogue to clarify the ship's maneuver; dialogue absent from the film because the maneuver itself can be shown. The information that *is* narrated is the *discourse*—a time-ordered, intentionally-selected subset of the plot (Genette 1980).

Algorithmically telling a story akin to Figure 1 demands formalizing plot, discourse, and narration. This is often not practically feasible, which invites the choice of which layers to model and which others to control for. The choice ultimately depends on *authorial intent* (Pratt 1977)—the communicative goals (e.g., entertain, teach, inspire) to accomplish via the telling of the story. Above, the authors entertain the audience by carefully preserving ambiguity to build suspense until the last panel reveals a surprise: the crew's "safe haven" is in fact a deadly threat.

## 2.2 Narrative as a Designed Artifact

Telling a story is a *design task* (Simon 1996). Methods to accomplish design tasks are not strictly "right" or "wrong," but rather "better" or "worse" along *qualitative dimensions* that the designer cares about. In storytelling, quality is assessed relative to authorial intent. Thus, the quality of a story goes beyond just *what* is told. An author may also care about *how* it is told, *why* it is told, to *whom*, *when*, and *where*—all to varying, use-inspired degrees and extents.

Consider that in Figure 1, Han Solo's actions do not simply emerge from his rational deliberation as an agent in a task environment. While his behavior may accomplish the authorial goal of leading the Millennium Falcon's crew to safety, it is irrational w.r.t. the original plot-level intent of hiding from the Empire. Nonetheless, his actions are *authorially* planned for precisely because of their discourse-level narrative quality: they advance the plot in a deliberately *suspenseful* way.

Ultimately, a narrative is successful to the degree it accomplishes an author's intent. While similar to the process of anticipating a plan's execution dynamics (as in robotic task planning; e.g., Cashmore et al. 2019), it is further complicated by the nature of storytelling: the domain of discourse is different when narrative plans are narrated, as plan effects are manifest in the *mental states* of the audience. Thus, a narrative's success is not directly observable and must be empirically assessed. For our example, the comic would succeed as a narrative artifact if it predictably elicits entertaining suspense in its audiences.

## 2.3 The Narrative Planning Challenge

The central challenge for developing narrative planning systems is two-fold. The first challenge is computationally modeling both senses of narrative – as communicative act and as designed artifact – in terms of structural features of plans and their construction (search) processes. This could mean modeling narratives in such a way that classical planners can produce them; e.g., via compilation (Haslum 2012; Christensen, Nelson, and Cardona-Rivera 2020). More commonly, it means developing novel planners with expanded knowledge representations and reasoning mechanisms. We focus on this latter approach.

Importantly, the choice of modeling technique presumes a researcher has identified a desired *narrative quality*—i.e., the authorial intent that should be achieved by the narrative planner during its operation. This narrative quality defines the second challenge: the empirical assessment of how well a developed narrative planner predictably elicits (in an audience) the phenomenon it purports to model. In other words, the researcher must assess their system's success by answering the question: "to what degree does my narrative planner accomplish the authorial intent I care about?" The answer usually manifests as an experimental inquiry to determine how well a resulting narrative plan, its construction process, or both evokes the desired communicative goal in an audience. For example, given a plan-based representation of Figure 1, the SUSPENSER narrative planner (Cheong and Young 2014) predicts what discursive information would elicit suspense in an audience, as evidenced by several human-subjects studies.

## 3  Introduction to Narrative Planning

Narrative planning frames a planning agent as a *story director*. The story director must craft a *narrative plan* that represents a *narrative*—the product of a narration of a sequence of events that constitutes a trajectory through states-of-affairs. Like other planning agents, a story director crafts a plan by searching over a graph constructed from a declarative description of a task environment. In our case, the description is a *narrative problem*, isomorphic to a *classical problem*.

**Revisiting Classical Problems**  We briefly review the classical problem representation that most narrative planners rely on: STRIPS (Fikes and Nilsson 1971), where a problem is a tuple $P = \langle \mathcal{L}, I, A, G, f_{\text{cost}} \rangle$. $\mathcal{L}$ is a set of atomic well-formed formulae (wff) or their negations, composed from a formal language of predicate $\mathbb{P}$, constant $\mathbb{C}$, and variable $\mathbb{V}$ symbols (and no function symbols).[1] $I \subseteq \mathcal{L}$ is an initial state that obeys the closed-world assumption, $G \subseteq \mathcal{L}$ is a set of goal conditions, and $A$ is a set of actions.

An action is a state-transition represented by a triple $a = \langle \text{PRE}(a), \text{ADD}(a), \text{DEL}(a) \rangle$; respectively the precondition, add, and delete lists, all subsets of $\mathcal{L}$. An action $a$ is applicable in a state $s$ if $\text{PRE}(a) \subseteq s$. Applying it results in a state $s' = (s \backslash \text{DEL}(a)) \cup \text{ADD}(a)$ and incurs a cost per the function $f_{\text{cost}} \colon A \to \mathbb{R}^{0+}$; we assume $f_{\text{cost}} = 1$.

The solution to $P$ is a classical plan $\pi = [a_1, ..., a_m]$. This ordered sequence of actions $a_i \in A$ transforms the problem's initial state $I$ to a state $s_m$ that satisfies the goal; *i.e.* $G \subseteq s_m$. Its cost is computed as $c(\pi) = \sum_{a_i \in \pi} f_{\text{cost}}(a_i)$.

### 3.1  Narrative Planning Problems and Solutions

A *narrative* (planning) *problem* is broadly defined using the same ingredients: $\mathcal{L}$, $I$, $A$, $G$, and $f_{\text{cost}}$. However, unlike its classical counterpart, a narrative problem represents a formal description of authorial intent (including the task environment with which to achieve it). It thus encodes the communicative goals that the researcher would like to accomplish via the narrative planner. These goals are the ones the story director *must* aim for during the construction of the narrative plan.

Critically, authorial intent encompasses more than just the *achievement* of conditions specified within $G$. The planner also tacitly encodes authorial intent via the *maintenance* of conditions *during plan construction* demanded by the researcher's desired quality. The narrative plan construction process is therefore key: it must guarantee that the solution $\pi$ exhibits the authorially-intended qualities when it (or some morphism of it) is narrated to an audience.

For example, the authorial intent of the PROVANT narrative planner (Porteous and Lindsay 2019) is the generation of narrative that exhibits a canonical Hollywood-style struggle between a protagonist and antagonist. This is guaranteed by more than just the statement of conditions to achieve in $G$: it is *also* guaranteed by its plan construction process. During operation, PROVANT rules out portions of the search space that do not conform to the canonical Hollywood form. To expand, the comic in Figure 1 would not be an output reachable by PROVANT, as it does not exhibit the desired authorial intent.

A consequence of what narrative problems mean is that (by default) classical planners are insufficient for storytelling.[2] To illustrate, we reconsider Han's apparent irrationality: his actions are not what one might expect from (for example) a cooperative multi-agent belief-desire-intention (BDI) planning architecture (Rao and Georgeff 1995). In fact, his actions might never be considered for inclusion in $\pi$ because they potentially thwart the intent of all other agents. What we need instead is the ability to reason about how elements added to $\pi$ achieve authorial intent, sometimes to the apparent detriment of the characters within $\pi$.

The preceding considerations explain why classical notions of plan quality (e.g., cost, length) are insufficient: the anticipated audience's reception of the narrative plan must be a critical part of a story director's plan quality assessment. Han's actions are deliberately chosen because of their expected value to the *tellability* of the resulting narrative.

### 3.2  Points of Divergence in Narrative Planning

As mentioned, researchers typically focus on dimensions of narrative plan quality they care about.

For instance, a great deal of recent work has modeled agents who obtain mistaken beliefs and act per them such that they fail. These failed actions, like the disparities of belief used to prompt them, can be used to create irony as in the SABRE planner (Shirvani, Ware, and Farrell 2017), support the illusion of theory of mind as in the IMPRACTICAL planner (Teutenberg and Porteous 2013), and build tension as in the HEADSPACE planner (Sanghrajka, Young, and Thorne 2022). Of these, HEADSPACE *would* be able to approximate generating narratives like our Figure 1: the crew's mistaken beliefs are what lead them inside the giant space worm to begin with.

---

[1]Words in `true-type` font are predicates. When prepended with '?', they are variables. When `TitleCase`, they are constants.

[2]The storytelling limitations of classical planners are not simply addressed by non-classical (e.g., conformant) approaches.

The choice of which phenomenon to model is the major point of divergence within the narrative planning community. Different researchers seek to model different narrative-theoretic dimensions, each with unique rationales that explain why a given dimension matters for narrative plan quality. Presently, there is no consensus on which dimensions matter the *most*. We further contend: if we accept that storytelling is a design task, then there is no universal set of narrative-theoretic dimensions that are *sufficient* to characterize stories. Transitively, there is a rich (potentially infinite) set of storytelling forms and phenomena that researchers may reason about via planning systems.

Reasoning about diverse narrative phenomena requires adapting the narrative problem representation, the narrative planning process, or – more commonly – both. In the cited examples, the problem representation is expanded in different ways. For example, in HEADSPACE the wff in $\mathcal{L}$, $I$, and $G$ are expanded to admit statements about positive and negative character beliefs, and actions in $A$ must specify belief-based PRE(a), ADD(a), DEL(a) lists. This is like IMPRACTICAL, which further distinguishes conscious(?char) and at(?char, ?location) in $\mathcal{L}$ to reason about characters who witness others doing things at given locations, and transitively, characters who can predict when others do so to afford generating stories in which characters deceive one another.

### 3.3 Points of Convergence in Narrative Planning

While the diverse motivations for modeling narratives as plans has precluded a standard specification of narrative problems, the narrative planning community has tacitly converged on several "fundamental particles." These include characters, locations, and actions as distinguished plot concepts that are privileged in human cognition as these are the basis for *mental models*—our mental simulation of possible worlds—and *event models*—our mental simulation of sequences of events. Both models underlie our ability to make sense of stories (Cardona-Rivera and Young 2019). We describe other key points of convergence below.

Several narrative planners distinguish *outcome* $G_\text{o}$ from *trajectory* $G_t$ goals within $G$. The former are akin to classical planning ones: desired outcomes for the story's end. The latter are most similar to state-space planning *landmarks* (Hoffmann, Porteous, and Sebastia 2004) or plan-space planning *islands* (Hayes-Roth and Hayes-Roth 1979). Whereas landmarks or islands provide *guidance* to non-narrative planners, narrative problems admit them to afford authors more-direct expression of authorial intent over solution narrative plans (Riedl 2009).

Several narrative planners further partition $A$ into two narratological classes. *Happenings* $A_\text{h}$ are actions that can occur without reason, *e.g.* an accident. *Non-happenings*, on the other hand, must be intended (Bratman 1987)—they are carried out by characters in service of goals they have adopted, and are termed *intentional* actions $A_\text{I}$. A happening is isomorphic to a STRIPS-style action, whereas an intentional action is a quadruple $a = \langle \text{PRE}(a), \text{ADD}(a), \text{DEL}(a), \text{CHA}(a)\rangle$, where PRE$(a)$, ADD$(a)$, and DEL$(a)$ are as before and CHA$(a)$ is a list of terms in $\mathbb{C}$ that denote plot characters.

Another point of broad convergence is the role of a narrative plan's *causal coherence*, that each action that takes place has its preconditions satisfied. This causal backbone is critical for audiences to derive *temporal* sequences—people cannot easily understand stories without a spatio-temporal frame (Radvansky and Zacks 2014). A narrative plan's causal coherence contributes the tacitly-valued quality of *comprehensibility*, that the resulting narrative can be understood by audiences. Several studies have demonstrated that causally-coherent narrative plans rendered as textual or filmic media can themselves be used to predict average human answers to comprehension questions about the plan's constituent actions (Christian and Young 2004; Cardona-Rivera et al. 2016). For a given action $a_i \in \pi$, we can predict people's responses to the questions *Why / How did $a_i$ happen?*, *What enabled $a_i$ to happen?*, and *What was the consequence of $a_i$?*

The last point of broad convergence concerns another tacitly-valued dimension of narrative plan quality: *believability*, that the resulting narrative does not thwart the audience's *willing suspension of disbelief* or sense of being in a fictional world (Holland 2003). Believability is what motivated partitioning $A$ into $A_\text{h}$ and $A_\text{I}$ in the first place. This partition is needed because *ceteris paribus* a story director will search to satisfy $G$, without regard to whether all $a_i \in \pi$ are believable for characters to execute. For example, suppose our narrative problem describes part of our comic, with Leia and Han, aboard the Millennium Falcon. Further suppose the author specifies $G_\text{o} = \{\textbf{not}\ (\texttt{conscious(Han)})\}$. A plausible narrative plan $\pi$ narrated via templated text is:

(1) Leia picks up a blaster. (2) Leia stuns Han.

This plan is causally coherent; *e.g.* the blaster must be picked up in order to stun Han. However, is this plan believable as a story? Perhaps, but nothing in $\pi$ would structurally justify that. Why Leia strikes Han is not clear. Arguably, we need more context to understand how Leia's actions are believable. In other words, the plan lacks *motivational coherence*: the plan contains actions that do not appear to be motivated by anything. The IPOCL planner was developed to address this concern (Riedl and Young 2010), under the rationale that *intentions* are a distinguishing feature of anthropomorphic activity (Bates 1994; Dennett 1989) and that intentions provide motive to act (Bratman 1987). IPOCL expanded the wff in $\mathcal{L}$, $I$, and $G$ to admit statements about character intentions. These afford the expression of *character* goals as modal sentences of the form intends($c$, $g_c$), where $c$ represents any term from $\mathbb{C}$ that denotes a plot character and $g_c \in \mathcal{L}$ represents a condition the character $c$ intends to accomplish. These intentions constrain the search space of the story director: an action $a \in A_\text{I}$ can only be considered for expanding the search space if it is possible to make $a$ part of a character $c$'s sub-plan $\pi_{g_c} \subseteq \pi$ that accomplishes $g_c$. Today, narrative planners largely take intentional actions as a given,[3] as we do for the remainder of this paper.

---

[3]State-of-the-art planners such as HEADSPACE, SABRE, and IMPRACTICAL all reason about character intentions as sub-parts of their primary modeling purpose.

## 4 The Different Layers of Narrative

Fully modeling a narrative as we have defined would require formalizing all layers we have discussed thus far: plot, discourse, and narration. This task is often not practically feasible, leading researchers to pragmatically choose *which* layers to model. Each layer re-casts the classical problem representation to mean different narrative concepts. Below, we review different framings and extract common themes.

### 4.1 Plot Planning

The bulk of the work in narrative planning has focused on modeling plot, in which the narrative plan is meant to represent the plot structure: a sequence of actions taken by characters in the story that evolve the virtual world from its initial configuration to an author-desired one. When representing plot, the narrative problem is similar to a multi-agent planning (MAP) one (Brafman and Domshlak 2013). This is because characters exhibit intention dynamics as they are orchestrated by the story director toward achieving $G$. But unlike MAP, it is plausible for characters to *intentionally* conflict, under the rationale that this phenomenon features prominently in global Western narratives (Herman, Manfred, and Ryan 2010). Whereas plans with resource or coordination conflicts would be ruled out in MAP, they are made possible by design in the GLAIVE planner (Ware and Young 2014): its search space allows the story director to find plans where characters adopt intentions that are mutually exclusive, relaxing the IPOCL requirement that a character $c$'s subplan $\pi_{g_c}$ necessarily achieve $g_c$. In other words, characters can take actions toward goals, but fail to accomplish them.

Plot-level plans ought to be sound with respect to the story world domain and problem in which they take place. But because these task environments are virtual, they can themselves be modified to suit particular storytelling goals. For instance, the INITIAL STATE REVISION (ISR) algorithm (Riedl and Young 2005) partitions $I$ into true, false, and undetermined sentence sets whose combinations determine the plot's set of *alternative possible worlds* (Ryan 1991). While this is similar to $I$ being an open-world state, ISR shifts undetermined sentences into true or false as convenient to accomplish storytelling goals, which only makes sense per the synthetic nature of the plot's virtual world.

Because quality is assessed with respect to an audience, plot plans must be narrated in some way in order to empirically evaluate whether the plot model achieves its intended effect. Today, most narrative planners tacitly follow Reiter and Dale's (2000) Natural Language Generation (NLG) pipeline, which starts with (plot) *Content Determination*, is followed by (discourse) *Content Structuring*, and ends with (narration) *Linguistic Realization*. Plot planners typically have a perfunctory NLG pipeline, to control for any spurious discourse and narration effects on comprehension due to natural language. But because AI planning is itself an effective model of NLG, a separate line of research has focused on narrative discourse planning.

### 4.2 (Narrative) Discourse Planning

As in conventional discourse planning, narrative plans are meant to reflect the informational and intentional structure of a discourse (Grosz and Sidner 1986), within a sequence of *communicative* actions taken by the story director that are intended to evolve the mental state of the audience from its (initial) state prior to experiencing the discourse to an author-desired one. When representing discourse, the narrative problem is similar to the more general discourse planning one (Garoufi 2014). This is because the story director relies on the framing of utterances as *speech actions* and treats communication as a goal-oriented process in the space of audience beliefs (Cohen and Perrault 1979). But unlike its more general formulation (cf. Gatt and Krahmer, 2018), narrative discourse planning does not assume that the beliefs of the audience monotonically increase over time. Nor does it assume that the goal must be a belief that the audience should obtain at the *end* of the discourse. In fact, the audience's *belief dynamics* – i.e., the trajectory of belief expansion, contraction, and revision operations (cf. Alchourrón, Gärdenfors, and Makinson, 1985) – is a key determinant of *narrative coherence*; i.e., that the artifact is received *as* a narrative (Herman 2013). This quality is particularly important in human-AI applications: information parsed by people as a story is better comprehended and better retained by them relative to non-story information (Fisher and Radvansky 2018).

In the pipeline approach, the input plot plan $\pi_{\text{plot}}$ represents the knowledge base and informs the material communicative goals that the narrative discourse planner will strive for. Thus, elements of $\pi_{\text{plot}}$ remain fixed and are reified: they become part of the language $\mathcal{L}$ for the narrative *discourse* problem. Narrative discourse planning operates in belief space, and thus $\mathcal{L}$ also admits statements about belief—these are used to specify the expected audience's mental state before narration $I$, the author-desired mental state outcome for the plot's telling $G$, and the communication actions $A$ that will effect changes in audience belief.

Thus, for narrative discourse planning, the actions in $A$ tacitly specify a model of *belief-based* narrative-theoretic phenomena: they manipulate the presentation of plot details to elicit particular belief dynamics in audiences, which in turn result in particular narrative effects. Examples include the use of *staging* via MISER (Matthews et al. 2017) and elicitation of *inferencing* via INFER (Niehaus and Young 2014).

### 4.3 Narration Planning

Typically, narrative discourse planners are tightly-coupled to the narration, such that the solution narrative discourse plan is itself the realization of the plot in a given medium. As a result, narrative-theoretic plan-based linguistic realization is relatively under-explored; the FIREBOLT cinematic realization planner (Thorne et al. 2019) is a notable exception.

Other work in this area has sought to break-away from the NLG pipeline, under the rationale that the storytelling's *form* (i.e., discourse and narration) cannot be separate from its (plot) *content* (Elliott 2004). Work includes adapting the direction of the pipeline via specification of plot-level landmarks based on required discourse "snapshots" as in PLOTSHOT (Cardona-Rivera and Li 2016), as well as doing away with the pipeline altogether in order to co-evolve plot and discourse (coupled to narration) simultaneously as in BIPOCL (Winer and Young 2016).

## 5 Three Key Narrative Planning Paradigms

Techniques to solve narrative problems are tightly-coupled to a problem representation—the one needed to formalize the particular class of narrative phenomena of interest to the modeler. A fundamental assumption they share is that storytelling is well-modeled as a search process.

At the same time, different planning paradigms afford *different* ways of modeling this search process and transitively, how to *think* about modeling narrative phenomena. Different paradigms offer different spaces that shape the story director's range of generatable stories, or *expressive range* (Summerville 2018). We cover three broadly-used paradigms.

### 5.1 Modeling Narrative via Plan-space Planning

Plan-space search operates over a graph in which nodes represent partial-plans and arcs represent plan refinement operations (Kambhampati, Knoblock, and Yang 1995). For example, in POCL planning (Weld 1994), refinements are introduced to guarantee that (1) no preconditions remain unsatisfied (recorded via *causal links*), and (2) no action in the plan could be ordered such that it threatens to undo (establish the opposite condition of) a causal link between two other actions. These respectively are *Open Condition* (OC) and *Threatened Link* (TL) flaws.

Plan-space narrative planning affords modeling story phenomena in terms of *narrative-theoretic* plan construction flaws and fixes. That is, a modeler must add (to OC and TL) new classes of flaws and fixes that in some way capture a dimension of plan quality with narrative import. For example, IPOCL introduced three flaws to model the intention dynamics that make character actions more believable when narrated. When IPOCL adds an action $a \in A_I$ to fix some other flaw, it must guarantee that $a$ is at some point added to a sub-plan $\pi_{g_c} \subseteq \pi$ that accomplishes $g_c$, $\forall c \in \text{CHA}(a)$; while $a$ remains un-added, the partial-plan (under refinement) has an *Unknown Intent* (UI) flaw. If any sub-plan $\pi_{g_c}$ is not preceded by an action that establishes the effect intends($c$, $g_c$) for the sub-plan's $c \in \mathbb{C}$, the partial-plan has an *Open Motivation* (OM) flaw, fixable by adding such an action. And if any two sub-plans $\pi^1_{g_c}, \pi^2_{g_c} \subseteq \pi$ assert opposite sub-goals (i.e., $g_c{}^1 = \neg g_c{}^2$) the partial-plan has a *Threatened Intent* (TI) flaw, fixable by ordering $\pi^1_{g_c}$ after $\pi^2_{g_c}$ or *vice-versa*.

This modeling strategy is attractive in that it directly shapes the underlying search space in a way that facilitates providing theoretical guarantees about the space of solutions. Flaws and their fixes respectively identify partial-plans that would not be solutions to a given narrative-theoretic problem class, and the algorithmic means to refine partial-plans such that they *do* become solutions (or fail in the attempt). The full modeling strategy then is to empirically demonstrate that narrative plans with the structural quality to-be-preserved (via flaw detection and refinement) do in fact elicit a particular narrative-theoretic effect of interest to the modeler. The drawback of plan-space narrative planners lies in their performance, but plan-space heuristics which seek to offset that penalty, such as those codified by VHPOP (Younes and Simmons 2003), give insights into potential directions.

### 5.2 Modeling Narrative via Hierarchical Planning

Hierarchical formalisms are varied, but broadly share the property of being more expressive and complex than classical planning: STRIPS-style *primitive* actions are complemented with isomorphic more-abstract *compound* actions that require *decomposition*, or associated sub-plan (Bercher, Alford, and Höller 2019). Decomposition happens via *methods* representing sub-goals that require sub-plans, to be further decomposed until all compound actions are reduced to primitives. The idea is that a decomposition method $d = \langle a_C, \pi_{\subseteq} \rangle$ maps a compound action $a_C$ to a sub-plan $\pi_{\subseteq}$ that depends on (i.e., has preconditions that relate to) $\text{PRE}(a_C)$ and contributes to (i.e., has effects that relate to) $\text{EFF}(a_C) = \text{ADD}(a_C) \cup \text{DEL}(a_C)$ (Bercher et al. 2016).

Hierarchical narrative planning affords modeling story phenomena in terms of narrative-theoretic recursive specifications, akin to grammar rewriting rules. This is well-suited for representing a wide variety of story phenomena that depend on abstraction, across the narrative layers. In plot, for example, character intentions can be straightforwardly codified (Cavazza, Charles, and Mead 2002): a compound action $a_C$ may assert intends($c$, $g_c$) $\in$ ADD($a_C$) with its decomposition being the sub-plan $\pi_{g_c}$ that achieves it. Another example is DARSHAK (Jhala and Young 2010), which uses hierarchies to bridge cinematic discourse and narration: a compound action's effects $\text{EFF}(a_C)$ represent discourse-layer information about the plot being filmed, and a method represents a *cinematic idiom* that identifies a narration-layer sub-plan of film shots thought to achieve $\text{EFF}(a_C)$.

Modeling via hierarchical formalisms is attractive due its potential for *authorial leverage* (Chen, Nelson, and Mateas 2009): that is, hierarchical story directors afford users significant power to define narrative plan quality aligned to their authorial intent, across all narrative layers. Like in other hierarchical planners, authors may introduce elements of "advice" to a hierarchical story director: decomposition methods afford ways to encode typical (not necessarily optimal) action sequences or *scripts* (Schank and Abelson 1975). For example, this is used in the GROUND DECOMPOSITIONAL PARTIAL ORDER PLANNER to suggest idiomatic film edits during the construction of a cinematic narration plan (Winer and Cardona-Rivera 2018). We observe that certain aspects of hierarchical representation provide a mechanism to exercise this authorial leverage via specification of non-classical temporally extended goals. For example, partially ordered compound tasks that are used to scaffold the structure of novel variants of the TV drama *Friends* (Cavazza, Charles, and Mead 2002). Related approaches to such non-classical goal specification are discussed below in the context of heuristic search planning.

### 5.3 Modeling Narrative via Heuristic Search

Heuristic search planning, the dominant current approach, plans via forward search through state space, evaluating states on the basis of general, domain-independent heuristics (Bonet and Geffner 2001). The narrative planning challenge is how best to encode narrative-theoretic phenomena, in order to leverage the efficient performance of such approaches.

One strategy has been to encode narrative phenomena as constraints within the narrative domain or as control knowledge to scaffold the structure of narrative. In the NETWORK-ING plan-based system, the use of constraints scaffolds construction of story genre-consistent sub-plans that enable characters to realize their intentions (Porteous, Charles, and Cavazza 2013). It has also featured as control knowledge to guide story development, as with the *affective* storytelling in the MADAME BOVARY system (Pizzi and Cavazza 2007). Such approaches have been shown to work well in practice, but they fail to capture more general narrative-theoretic phenomena.

Hence, other work has looked to develop *heuristics* that directly reason about narrative-theoretic properties. This is the case for GLAIVE (Ware and Young 2014), a forward search planner which generates motivationally coherent narrative plans, as discussed earlier. GLAIVE introduces a heuristic that incorporates character *intentions*, via a goal graph, which is used with an FF-style plan graph (Hoffmann and Nebel 2001) to calculate the heuristic estimate during search.

## 6 Open Challenges

While many open challenges are common to both narrative and non-narrative planning, others are driven by the differentiation required for plans to serve as narrative artifacts. We mention here two of the most significant: expanding expressive range and incorporating cognitive-driven generation.

**Increased Expressive Range**  There is a need to increase the expressive range of planning systems meant to create plot or narrative discourse structures. Narrative planners' expressive range – the breadth of structural features of their output artifacts – must differ from those of conventional planning because the structural features of narrative impose a set of distinct properties not readily accountable for by conventional planning knowledge representations. For example, plot structures contain actions that fail, explicit conflict between the actions and goals of their agents, and controlled increases in potential but unrealized threats to a plan's execution. Narrative discourse elements may work to intentionally obfuscate aspects of their domain of discourse from a reader/viewer. They may intentionally drive false beliefs or they may carefully curate and promote uncertainty, prompt repeated belief revision around specific concepts, play off of the relationships between the time of story events and the time of their telling, and focus as much on the trajectory of cognitive and emotional states during the experience of a reader as the set of beliefs they hold at the end of a narrative experience.

**Cognitive-driven Narrative Generation**  A second significant challenge for narrative planning is the need to strengthen our knowledge of the connection between narrative generation and narrative comprehension. In the field of cognitive psychology, narrative comprehension has long been studied by building cognitive models of narrative comprehension (Graesser and Franklin 1990) that posit that human readers progressively construct a mental concept graph during narrative comprehension that represents events in a story and their causal, temporal and spatial relationships, character intentions, and other elements with parallels to the contents of plan graphs. So similar are the parallels between representations in the two fields that narrative planning researchers have adapted the experimental methods used to validate these cognitive models for use by psychologists in evaluating the narrative coherence of generated plans (Riedl and Young 2010), but little work has been done to directly take in to account comprehension during narrative generation.

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
