# OpenReview forum: "The Story So Far on Narrative Planning"
_icaps-conference.org/ICAPS/2024/Conference — ICAPS 2024_

### Official Review · Reviewer_pmtL · 2024-01-02

**Significance And Importance:** 3
**Soundness:** 3
**Novelty:** 2
**Clarity:** 4
**Overall Evaluation:** 2
**Confidence:** 2

**Weaknesses:**

1: Minor weaknesses that are easily fixable.

**Contributions Of The Paper:**

The paper presents a survey on narrative planning, going from the problem definition to the different formalisms and methodologies to generate narratives, and finishing with a discussion on open challenges to the AI community.

**Ethical Considerations:**

(1) Not Applicable: The paper does not have any ethical considerations to address

**Nomination For Best Paper:**

No

**Questions For Authors:**

[Q1]. Could you add a discussion to the paper talking about the advantages and disadvantages (limitations?) of compiling narrative planning to (classical, temporal) planning?

**Reproducibility:**

0: N/A - nothing to reproduce.

**Strengths Of The Paper:**

The paper is very well written and makes for an easy and informative read. The structure of the paper helps non-experts like me to fully understand the different components of narrative planning, as well as the state of the art on this topic.

**Weaknesses Of The Paper:**

In Section 2.3, the survey mentions that there exist two main approaches to modeling and solving narrative planning tasks: either through compilations to classical planning or through developing novel planners with expanded knowledge representations and reasoning mechanisms. From that point, the survey solely focus on the latter, without further mentions to the compilation approach, which in my opinion offers some obvious benefits, i.e., a common description language (PDDL), a large number of solvers, heuristics etc. In fact, (Haslum 2012) shows how some compilations are more efficient than dedicated solvers for this task.
I can see how not all narrative planning objectives can be compiled to classical planning, how more expressive languages might be needed, or that domain-dependent heuristics and pruning techniques might be more efficient that domain-independent versions in some cases. I think the survey would highly benefit from such analysis: advantages and disadvantages (limitations?) of a compilation to (classical, temporal) planning. The current version of the paper spans 6.5/8 pages, so I think there is plenty of space to present this discussion.

[AFTER REBUTTAL] Thanks for answering my questions. I believe the paper will benefit from such section and discussion in the extra space.

---

> ### Author Rebuttal · Authors · 2024-01-26
>
> We thank you for your feedback about our paper, and for your remark calling attention to the compilation approach. To be candid, we originally had written a section on the compilation approach but cut it to fit within the allowed length for papers (discussed in the call). What we failed to notice is that ICAPS 2024 made special mention of survey papers being eligible for an expanded paper length to accommodate the references. We contacted the program chairs to confirm our eligibility, and they have confirmed we will have more space to address your concerns in good faith. We have thus added a section on compilation, in light of your concern.
>
> To anticipate the eventual section:
> Indeed, Haslum introduced two individual compilations that targeted motivational coherence, as first tackled by IPOCL. The "Meta Planning" compilation is less relevant: it is not domain-independent, and is not guaranteed to produce a semantically equivalent representation (as in, if a classical planner solves the compiled problem, the resulting plan is not guaranteed to be motivationally coherent in the IPOCL sense). Even Haslum notes this, with the caveat "although most of the time they will be" (with no indication of how often).
>
> The "Justification Tracking" compilation, on the other hand, is guaranteed to be semantically equivalent, but at a price: there is an exponential blowup in the problem size, in terms of 3 factors: the number of potential character intentions, the number of relevant intentional actions toward satisfying those intentions, and the number of characters needed for those intentional actions.  Haslum reported a 3 orders of magnitude speedup on the single problem used to originally benchmark IPOCL. However, Teutenberg and Porteous (2013) demonstrated that this speedup is not always guaranteed. In fact, relative to FF applied to this compilation, IMPRACTICAL is 2 orders of magnitude faster, and GLAIVE is 3 orders of magnitude faster.
>
> To our knowledge there is only one other approach that has tried compilation, but it compiles a more complex narrative problem to a simpler narrative problem, which ultimately is still solved by the Glaive narrative planner.
>
> Thus, in general, the purpose of compilation for narrative is less on the modeling of narrative phenomena per se, and more for the benefit of tractably solving narrative problems.
>
> Teutenberg and Porteous (2013). "Efficient intent-based narrative generation using multiple planning agents." Proceedings of AAMAS.

---

### Official Review · Reviewer_KsGB · 2024-01-09

**Significance And Importance:** 3
**Soundness:** 3
**Novelty:** 3
**Clarity:** 3
**Overall Evaluation:** 2
**Confidence:** 3

**Weaknesses:**

1: Minor weaknesses that are easily fixable.

**Contributions Of The Paper:**

This is a "position paper" and not a "technical paper", I believe. It provides a broad overview over work that has been done in the area of narrative planning and gives some hints where input from planning research could help. For the ICAPS audience, this paper is definitely an interesting piece of inspiration.

**Ethical Considerations:**

(1) Not Applicable: The paper does not have any ethical considerations to address

**Nomination For Best Paper:**

No

**Questions For Authors:**

1) How much theory of mind reasoning should be done by first principles?
2) Do you agree that while in automated planning, we just care for generating plans, in plot planning, also cognitive restrictions for generating plans should play a role?
3) Do you think that generating suspense is (just) a matter of creating possible risky situations and leaving open the concrete setting?

**Reproducibility:**

0: N/A - nothing to reproduce.

**Strengths Of The Paper:**

- It gives a good overview of the field of narrative planning.
- It lays out the different planning problems that you face in this context.
- It provides a number of interesting examples.
- It contains a lot of pointers to work that has been done in this area.
- In general, I like the paper because it highlights an application area that I have not seen at ICAPS too often.

**Weaknesses Of The Paper:**

I like Section 4, pointing out the different levels of planning. However, I missed a tight and formal interaction between the levels. For example, I could see here that narration planning could set objectives (such as open interpretations/ambiguities) based on narration goals such as suspense. In other words, I see here a potential two-level planning process, where one planning level is the object level of another planning process.

In Section 3.3, I was surprised not to see a reference to time-extended and trajectory goals using LTL and LTL_f as discussed recently in the planning community.

Similarly, I would have expected to see a reference to epistemic planning (see recent AIJ special issue on this topic), when it comes to Multi-Agent Planning. There is an interesting point here, that most often epistemic planning is cooperative. However, there are notable exceptions, such as the work by Andreas Witzel (in particular, his Ph.D. thesis).

In Section 5, I agree with the authors that there is an eminent difference between classical and hierarchical planning, and most probably the latter has advantages in the setting of narrative planning. However, I do not agree with the authors that the type of search should be taken into account. As a KR purist, I would focus on the representation, not on the processes.

More generally, I would have loved to see more concrete examples and more concrete problems to be solved by the planning community. But this is probably too much to ask for a conference paper. However, I would urge the authors of the paper to join up with ICAPS people and perhaps go for a Dagstuhl workshop on the topic of narrative planning, bringing both communities together.

---

> ### Author Rebuttal · Authors · 2024-01-26
>
> Thank you for your feedback, your grace in understanding how much we can fit into this paper, and for the references, which we will incorporate. Below we address the remaining critical weaknesses/questions.
>
> > "I see here a potential two-level planning process."
>
> Indeed, while we do not expound it in technical depth, this is precisely what BIPOCL does (see S.4.3)
>
> > "I was surprised not to see a reference to time-extended and trajectory goals using LTL and LTL_f"
>
> We too find it odd that temporal logics have not made more strides within narrative planning. We only know of one such work: TALplanner (Kvarnström and Doherty, 2000). We have updated our text accordingly.
>
> > "As a KR purist, I would focus on the representation, not on the processes."
>
> We understand this point; as McDermott espoused: "physics, not advice." Narrative planning needs to consider both KR and search, to guarantee that plans that do not satisfy narrative constraints are pruned. In hierarchical approaches, one could (e.g.) use this to afford expressing a preference over how to construct camera shots to represent artistic style. If certain camera shots are not the most expedient way to depict a scene, we could use decompositions to ensure that artistic styles are not inadvertently pruned by a cost-optimizing planner (as the mentioned GDPOP does).
>
> > "2) Do you agree that while in automated planning, we just care for generating plans, in plot planning, also cognitive restrictions for generating plans should play a role?"
>
> We agree that in the generation of plots, there are cognitive constraints that play a role (and are often leveraged by authors/planners). By analogy, cognitive constraints of our audience in navigating a belief space through limited observations provided at each communicative plan execution step are our equivalent to physical constraints in planning for robots.
>
> > "3) Do you think that generating suspense is (just) a matter of creating possible risky situations and leaving open the concrete setting?"
>
> Generating suspense is more nuanced than just creating possible risky situations: it depends on disparity of knowledge between the audience and the planner's KR. Branigan (1987) suggests suspense in film occurs whenever the audience and protagonist have a disparity of knowledge. The audience might know more about the world (such as an impending threat) than the protagonist or the narrator does not reveal information held by a protagonist about the world to the audience.

---

### Official Review · Reviewer_N1K1 · 2024-01-21

**Significance And Importance:** 2
**Soundness:** 4
**Novelty:** 2
**Clarity:** 4
**Overall Evaluation:** 3
**Confidence:** 3

**Weaknesses:**

2: No major or minor weaknesses.

**Contributions Of The Paper:**

The authors provide us with a very well-written review of the narrative planning problem and survey of the various automated planning approaches that are used in the field. This includes thorough, precise and concise explanations of terminology and a brief summary of open challenges which highlights very interesting problems. The topic is relevant to human-centred AI and its application areas, including social assistive agents in both health care and education for example. An educational and enjoyable read — thank you.

**Ethical Considerations:**

(1) Not Applicable: The paper does not have any ethical considerations to address

**Nomination For Best Paper:**

No

**Questions For Authors:**

In Section 6, you state “Narrative discourse elements may work to intentionally obfuscate aspects of their domain of discourse from their reader/viewer. …drive false belief revision around specific concepts…”.

Out of curiosity, and given that you explicitly mention assistive technologies in your introduction, are there assistive scenarios in healthcare or education where this may also hold and are there ethical considerations in those settings that need to be addressed? If so, what do the mechanisms in narrative planning look like such that they generate plans that are  "safe by design".

POST-REBUTTAL Thank you for addressing my question. I am sure that the updated manuscript will be of great interest and benefit to the community and I look forward, in future, to reading about the developments to the issues that you mention in the updated Open Challenges section.

**Reproducibility:**

0: N/A - nothing to reproduce.

**Strengths Of The Paper:**

This is a beautifully written, clear and well-structured paper that is highly informative and while at first glance, the length of the references section seems too long, this is certainly understandable (and even expected) given the aim of the paper. The example used throughout the paper perfectly and succinctly conveys the terminology and ideas of the field. In the era of LLMs, the use of automated planning “as a formal, rigorous and common vocabulary for framing advances in the field” is most welcome.

**Weaknesses Of The Paper:**

-

---

> ### Author Rebuttal · Authors · 2024-01-26
>
> We thank you for your review, for your kind words about our paper, and especially for your thoughtful question, pasted below:
>
> >> ... "Narrative discourse elements may work to intentionally obfuscate aspects of their domain of discourse..."
> > Out of curiosity, and given that you explicitly mention assistive technologies in your introduction, are there assistive scenarios in healthcare or education where this may also hold and are there ethical considerations in those settings that need to be addressed? If so, what do the mechanisms in narrative planning look like such that they generate plans that are  "safe by design".
>
> We agree with your observation here that narrative discourse elements and idioms may be used for obfuscation, filtering, and misrepresentation of information. We also agree this could be damaging in education and assistive technology deployments; we further foresee potential harms in the use of narrative planning for disinformation campaigns, radicalization narratives, and the computational codification of stereotype knowledge that general stories also suffer from. We thus sincerely appreciate your call to attention regarding ethical concerns.
>
> To concretely address these from a "safe by construction" approach: we believe that creating plan-based user models that simulate how a specific group of users might interpret the output of narrative planners may help mitigate adverse effects. This could be algorithmically possible by simulating narrative comprehension during construction (considering we can already predict normative answers to 5W/1H questions about the story), and then using the predicted comprehension to prune plans of the search space that do not satisfy a "comprehension constraint:" i.e., some (to-be-discovered) metric that codifies how likely the intended comprehension of the story actually manifests in the user.
>
> We also believe that automated planning might be useful to structurally (i.e., in terms of domain, action, state, and heuristic representations) understand what makes harmful stories so effective, which may then equip us to be mindful/guard against them at a structural level. We have expanded the Open Challenges section of our paper to include the above considerations, which form a really interesting basis for future work.

---

### Meta-Review · Area_Chair_LYfy · 2024-02-02

**Recommendation:** Accept (Oral)
**Confidence:** 4

**Metareview:**

All reviewers are clearly in favour of accepting the paper.

**Ethical Considerations:**

(1) Not Applicable: The paper does not have any ethical considerations to address